# Approximate Bayesian Image Interpretation using Generative Probabilistic Graphics Programs

Vikash K. Mansinghka* [1,2], Tejas D. Kulkarni* [1,2], Yura N. Perov[1,2,3], and Joshua B. Tenenbaum[1,2]

[1]Computer Science and Artificial Intelligence Laboratory, MIT
[2]Department of Brain and Cognitive Sciences, MIT
[3]Institute of Mathematics and Computer Science, Siberian Federal University

## Abstract

The idea of computer vision as the Bayesian inverse problem to computer graphics has a long history and an appealing elegance, but it has proved difficult to directly implement. Instead, most vision tasks are approached via complex bottom-up processing pipelines. Here we show that it is possible to write short, simple probabilistic graphics programs that define flexible generative models and to automatically them to interpret real-world images. Generative probabilistic graphics programs (GPGP) consist of a stochastic scene generator, a renderer based on graphics software, a stochastic likelihood model linking the renderer's output and the data, and latent variables that adjust the fidelity of the renderer and the tolerance of the likelihood. Representations and algorithms from computer graphics are used as the deterministic backbone for highly approximate and stochastic generative models. This formulation combines probabilistic programming, computer graphics, and approximate Bayesian computation, and depends only on general-purpose, automatic inference techniques. We describe two applications: reading sequences of degraded and adversarially obscured characters, and inferring 3D road models from vehicle-mounted camera images. Each of the probabilistic graphics programs we present relies on under 20 lines of probabilistic code, and yields accurate, approximately Bayesian inferences about real-world images.

## 1 Introduction

Computer vision has historically been formulated as the problem of producing symbolic descriptions of scenes from input images [10]. This is usually done by building bottom-up processing pipelines that isolate the portions of the image associated with each scene element and extract features that signal its identity. Many pattern recognition and learning techniques can then be used to build classifiers for individual scene elements, and sometimes to learn the features themselves [11, 7].

This approach has been remarkably successful, especially on problems of recognition. Bottom-up pipelines that combine image processing and machine learning can identify written characters with high accuracy and recognize objects from large sets of possibilities. However, the resulting systems typically require large training corpuses to achieve reasonable levels of accuracy, and are difficult both to build and modify. For example, the Tesseract system [16] for optical character recognition is over $10,000$ lines of C++. Small changes to the underlying assumptions frequently necessitates end-to-end retraining and/or redesign.

Generative models for a range of image parsing tasks are also being explored [17, 4, 18, 22, 20]. These provide an appealing avenue for integrating top-down constraints with bottom-up processing,

and provide an inspiration for the approach we take in this paper. But like traditional bottom-up pipelines for vision, these approaches have relied on considerable problem-specific engineering, chiefly to design and/or learn custom inference strategies, such as MCMC proposals [18, 22] that incorporate bottom-up cues. Other combinations of top-down knowledge with bottom up processing have been remarkably powerful [9]. For example, [8] has shown that global, 3D geometric information can significantly improve the performance of bottom-up object detectors.

In this paper, we propose a novel formulation of image interpretation problems, called generative probabilstic graphics programming (GPGP). GPGP shares a common template: a stochastic scene generator, an approximate renderer based on existing graphics software, a highly stochastic likelihood model for comparing the renderer's output with the observed data, and latent variables that control the fidelity of the renderer and the tolerance of the image likelihood. Our probabilistic graphics programs are written in Venture, a probabilistic programming language descended from Church [6]. Each model we introduce requires less than 20 lines of probabilistic code. The renderers and likelihoods for each are based on standard templates written as short Python programs. Unlike typical generative models for scene parsing, inverting our probabilistic graphics programs requires no custom inference algorithm design. Instead, we rely on the automatic Metropolis-Hastings (MH) transition operators provided by our probabilistic programming system. The approximations and stochasticity in our renderer, scene generator and likelihood models serve to implement a variant of approximate Bayesian computation [19, 12]. This combination can produce a kind of self-tuning analogue of annealing that facilities reliable convergence.

To the best of our knowledge, our GPGP framework is the first real-world image interpretation formulation to combine all of the following elements: probabilistic programming, automatic inference, computer graphics, and approximate Bayesian computation; this constitutes our main contribution. Our second contribution is to provide demonstrations of the efficacy of this approach on two image interpretation problems: reading snippets of degraded and adversarially obscured alphanumeric characters, and inferring 3D road models from vehicle mounted cameras. In both cases we quantitatively report the accuracy of our approach on representative test datasets, as compared to standard bottom-up baselines that have been extensively engineered.

## 2 Generative Probabilistic Graphics Programs and Approximate Bayesian Inference.

GPGP defines generative models for images by combining four components. The first is a *stochastic scene generator* written as probabilistic code that makes random choices for the location and configuration of the main elements in the scene. The second is an *approximate renderer based on existing graphics software* that maps a scene $S$ and control variables $X$ to an image $I_R = f(S, X)$. The third is a *stochastic likelihood model* for image data $I_D$ that enables scoring of rendered scenes given the control variables. The fourth is a set of latent variables $X$ that control the fidelity of the renderer and/or the tolerance in the stochastic likelihood model. These components are described schematically in Figure 1.

We formulate image interpretation tasks in terms of sampling (approximately) from the posterior distribution over images:

$$P(S|I_D) \propto \int P(S)P(X)\delta_{f(S,X)}(I_R)P(I_D|I_R, X)dX$$

We perform inference over execution histories of our probabilistic graphics programs using a uniform mixture of generic, single-variable Metropolis-Hastings transitions, without any custom, bottom-up proposals. We first give a general description of the generative model and inference algorithm induced by our probabilistic graphics programs; in later sections, we describe specific details for each application.

Let $S = \{S_i\}$ be a decomposition of the scene $S$ into parts $S_i$ with independent priors $P(S_i)$. For example, in our text application, the $S_i$s include binary indicators for the presence or absence of each glyph, along with its identity ("A" through "Z", plus digits 0-9), and parameters including location, size and rotation. Also let $X = \{X_j\}$ be a decomposition of the control variables $X$ into parts $X_j$ with priors $P(X_j)$, such as the bandwidths of per-glyph Gaussian spatial blur kernels, the variance

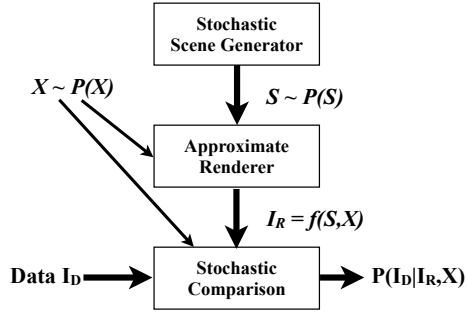

Figure 1: An overview of the GPGP framework. Each of our models shares a common template: a stochastic scene generator which samples possible scenes $S$ according to their prior, latent variables $X$ that control the fidelity of the rendering and the tolerance of the model, an approximate render $f(S, X) \rightarrow I_R$ based on existing graphics software, and a stochastic likelihood model $P(I_D|I_R, X)$ that links observed rendered images. A scene $S^*$ sampled from the scene generator according to $P(S)$ could be rendered onto a single image $I_R^*$. This would be extremely unlikely to exactly match the data $I_D^*$. Instead of requiring exact matches, our formulation can broaden the renderer's output $P(I_R|S*)$ and the image likelihood $P(I_D^*|I_R)$ via the latent control variables $X$. Inference over $X$ mediates the degree of smoothing in the posterior.

of a Gaussian image likelihood, and so on. Our proposals modify single elements of the scene and control variables at a time, as follows:

$$P(S) = \prod_i P(S_i) \qquad q_i(S_i', S_i) = P(S_i') \qquad P(X) = \prod_j P(X_j) \qquad q_j(X_j', X_j) = P(X_j')$$

Now let $K = |\{S_i\}| + |\{X_j\}|$ be the total number of random variables in each execution. For simplicity, we describe the case where this number can be bounded above beforehand, i.e. total a priori scene complexity is limited. At each inference step, we choose a random variable index $k < K$ uniformly at random. If $k$ corresponds to a scene variable $i$, then we propose from $q_i(S_i', S_i)$, so our overall proposal kernel $q((S, X) \rightarrow (S', X')) = \delta_{S_{-i}}(S')P(S_i')\delta_X(X')$. If $k$ corresponds to a control variable $j$, we propose from $q_j(X_j', X_j)$. In both cases we re-render the scene $I_R' = f(S', X')$. We then run the kernel associated with this variable, and accept or reject via the MH equation:

$$\alpha_{MH}((S, X) \rightarrow (S', X')) = min\Big(1, \frac{P(I_D|f(S', X'), X')P(S')P(X')q((S', X') \rightarrow (S, X))}{P(I_D|f(S, X), X)P(S)P(X)q((S, X) \rightarrow (S', X'))}\Big)$$

We implement our probabilistic graphics programs in the Venture probabilistic programming language. The Metropolis-Hastings inference algorithm we use is provided by default in this system; no custom inference code is required. In the context of our GPGP formulation, this algorithm makes implicit use of ideas from approximate Bayesian computation (ABC). ABC methods approximate Bayesian inference over complex generative processes by using an exogenous distance function to compare sampled outputs with observed data. In the original rejection sampling formulation, samples are accepted only if they match the data within a hard threshold. Subsequently, combinations of ABC and MCMC were proposed [12], including variants with inference over the threshold value [15]. Most recently, extensions have been introduced where the hard cutoff is replaced with a stochastic likelihood model [19]. Our formulation incorporates a combination of these insights: rendered scenes are only approximately constrained to match the observed image, with the tightness of the match mediated by inference over factors such as the fidelity of the rendering and the stochasticity in the likelihood. This allows image variability that is unnecessary or even undesirable to model to be treated in a principled fashion.

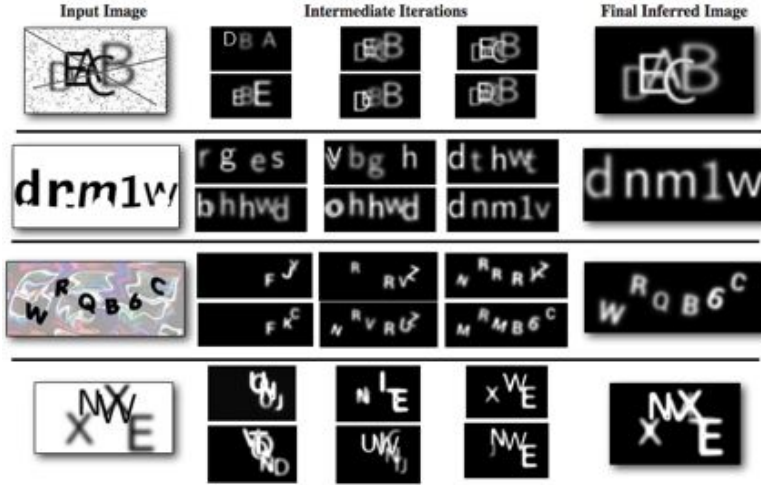

Figure 2: Four input images from our CAPTCHA corpus, along with the final results and convergence trajectory of typical inference runs. The first row is a highly cluttered synthetic CAPTCHA exhibiting extreme letter overlap. The second row is a CAPTCHA from TurboTax, the third row is a CAPTCHA from AOL, and the fourth row shows an example where our system makes errors on some runs. Our probabilistic graphics program did not originally support rotation, which was needed for the AOL CAPTCHAs; adding it required only 1 additional line of probabilistic code. See the main text for quantitative details, and supplemental material for the full corpus.

## 3    Generative Probabilistic Graphics in 2D for Reading Degraded Text.

We developed a probabilistic graphics program for reading short snippets of degraded text consisting of arbitrary digits and letters. See Figure 2 for representative inputs and outputs. In this program, the latent scene $S = \{S_i\}$ contains a bank of variables for each glyph, including whether a potential letter is present or absent from the scene, what its spatial coordinates and size are, what its identity is, and how it is rotated:

$$P(S_i^{\text{pres}} = 1) = 0.5 \quad P(S_i^x = x) = \begin{cases} 1/w & 0 \le x \le w \\ 0 & \text{otherwise} \end{cases} \quad P(S_i^y = y) = \begin{cases} 1/h & 0 \le x \le h \\ 0 & \text{otherwise} \end{cases}$$

$$P(S_i^{\text{glyph\_id}} = g) = \begin{cases} 1/G & 0 \le S_i^{\text{glyph\_id}} < G \\ 0 & \text{otherwise} \end{cases} \quad P(S_i^\theta = g) = \begin{cases} 1/2\theta^{\max} & -\theta^{\max} \le S_i^\theta < \theta^{\max} \\ 0 & \text{otherwise} \end{cases}$$

Our renderer rasterizes each letter independently, applies a spatial blur to each image, composites the letters, and then blurs the result. We also applied global blur to the original training image before applying the stochastic likelihood model on the blurred original and rendered images. The stochastic likelihood model is a multivariate Gaussian whose mean is the blurry rendering; formally, $I_D \sim N(I_R; \sigma)$. The control variables $X = \{X_j\}$ for the renderer and likelihood consist of per-letter Gaussian spatial blur bandwidths $X_j^i \sim \lambda \cdot Beta(1, 2)$, a global image blur on the rendered image $X_{\text{blur\_rendered}} \sim \beta \cdot Beta(1, 2)$, a global image blur on the original test image $X_{\text{blur\_test}} \sim \gamma \cdot Beta(1, 2)$, and the standard deviation of the Gaussian likelihood $\sigma \sim Gamma(1, 1)$ (with $\lambda$, $\gamma$ and $\beta$ set to favor small bandwidths). To make hard classification decisions, we use the sample with lowest pixel reconstruction error from a set of 5 approximate posterior samples. We also experimented with enabling enumerative (griddy) Gibbs sampling for uniform discrete variables with 10% probability. The probabilistic code for this model is shown in Figure 4.

To assess the accuracy of our approach on adversarially obscured text, we developed a corpus consisting of over 40 images from widely used websites such as TurboTax, E-Trade, and AOL, plus additional challenging synthetic CAPTCHAs with high degrees of letter overlap and superimposed distractors. Each source of text violates the underlying assumptions of our probabilistic graphics program in different ways. TurboTax CAPTCHAs incorporate occlusions that break strokes within

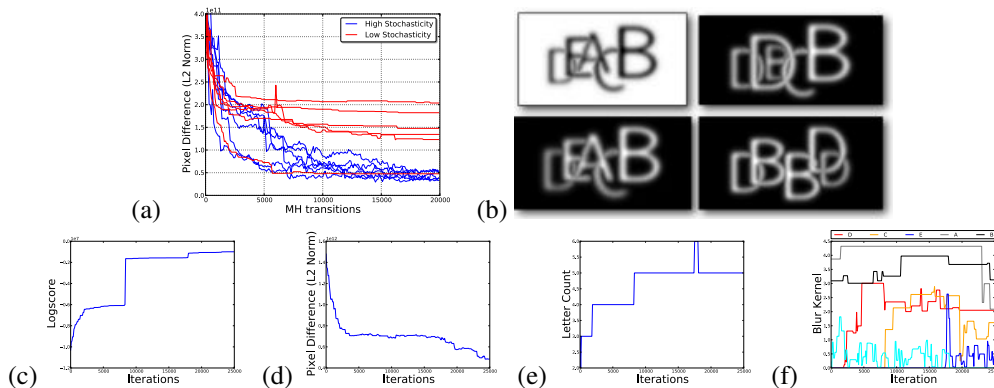

Figure 3: Inference over renderer fidelity significantly improves the reliability of inference. **(a)** Reconstruction errors for 5 runs of two variants of our probabilistic graphics program for text. Without sufficient stochasticity and approximation in the generative model — that is, with a strong prior over a purely deterministic, high-fidelity renderer — inference gets stuck in local energy minima (red lines). With inference over renderer fidelity via per-letter and global blur, the tolerance of the image likelihood, and the number of letters, convergence improves substantially (blue lines). Many local minima in the likelihood are escaped over the course of single-variable inference, and the blur variables are automatically adjusted to support localizing and identifying letters. **(b)** Clockwise from top left: an input CAPTCHA, two typical local minima, and one correct parse. **(c,d,e,f)** A representative run, illustrating the convergence dynamics that result from inference over the renderer's fidelity. From left to right, we show overall log probability, pixel-wise disagreement (many local minima are escaped over the course of inference), the number of active letters in the scene, and the per-letter blur variables. Inference automatically adjusts blur so that newly proposed letters are often blurred out until they are localized and identified accurately.

letters, while AOL CAPTCHAs include per-letter warping. These CAPTCHAs all involve arbitrary digits and letters, and as a result lack cues from word identity that the best published CAPTCHA breaking systems depend on [13]. The dynamically-adjustable fidelity of our approximate renderer and the high stochasticity of our generative model appear to be necessary for inference to robustly escape local minima. We have observed a kind of self-tuning annealing resulting from inference over the control variables; see Figure 3 for an illustration. We observe robust character recognition given enough inference, with an overall character detection rate of 70.6%. To calibrate the difficulty of our corpus, we also ran the Tesseract optical character recognition engine [16] on our corpus; its character detection rate was 37.7%.

## 4 Generative Probabilistic Graphics in 3D: Road Finding.

We have also developed a generative probabilistic graphics program for localizing roads in 3D from single images. This is an important problem in autonomous driving. As with many perception problems in robotics, there is clear scene structure to exploit, but also considerable uncertainty about the scene, as well as substantial image-to-image variability that needs to be robustly ignored. See Figure 5b for example inputs.

The probabilistic graphics program we use for this problem is shown in Figure 7. The latent scene $S$ is comprised of the height of the roadway from the ground plane, the road's width and lane size, and the 3D offset of the corner of the road from the (arbitrary) camera location. The prior encodes assumption that the lanes are small relative to the road, and that the road has two lanes and is very likely to be visible (but may not be centered). This scene is then rendered to produce a *surface-based segmentation image*, that assigns each input pixel to one of 4 regions $r \in R = \{\text{left offroad}, \text{right offroad}, \text{road}, \text{lane}\}$. Rendering is done for each scene element separately, followed by compositing, as with our 2D text program. See Figure 5a for random surface-based segmentation images drawn from this prior. Extensions to richer road and ground geometries are an interesting direction for future work. This model is similar in spirit to [1] but the key differ-

```
ASSUME is_present (mem (lambda (id) (bernoulli 0.5)))
ASSUME pos_x (mem (lambda (id) (uniform_discrete 0 200)))
ASSUME pos_y (mem (lambda (id) (uniform_discrete 0 200)))
ASSUME size_x (mem (lambda (id) (uniform_discrete 0 100)))
ASSUME size_y (mem (lambda (id) (uniform_discrete 0 100)))
ASSUME rotation (mem (lambda (id) (uniform_continuous -20.0 20.0)))
ASSUME glyph (mem (lambda (id) (uniform_discrete 0 35))) // 26 + 10.
ASSUME blur (mem (lambda (id) (* 7 (beta 1 2))))
ASSUME global_blur (* 7 (beta 1 2))
ASSUME data_blur (* 7 (beta 1 2))
ASSUME epsilon (gamma 1 1)
ASSUME data (load_image "captcha_1.png" data_blur)
ASSUME image (render_surfaces max-num-glyphs global_blur
(pos_x 1) (pos_y 1) (glyph 1) (size_x 1) (size_y 1) (rotation 1) (blur 1)
(is_present 1) (pos_x 2) (pos_y 2) (glyph 2) (size_x 2) (size_y 2)
(rotation 2) (blur 2) (is_present 2) ... (is_present 10))
OBSERVE (incorporate_stochastic_likelihood data image epsilon) True
```

Figure 4: A generative probabilistic graphics program for reading degraded text. The scene generator chooses letter identity (A-Z and digits 0-9), position, size and rotation at random. These random variables are fed into the renderer, along with the bandwidths of a series of spatial blur kernels (one per letter, another for the overall rendered image from generative model and another for the original input image). These blur kernels control the fidelity of the rendered image. The image returned by the renderer is compared to the data via a pixel-wise Gaussian likelihood model, whose variance is also an unknown variable.

ence is that our framework relies on automatic inference techniques, is representationally richer due to compact model description and goes beyond point estimates to report posterior uncertainty.

In our experiments, we used $k$-means (with $k = 20$) to cluster RGB values from a randomly chosen training image. We used these clusters to build a compact appearance model based on cluster-center histograms, by assigning text image pixels to their nearest cluster. However, we are agnostic to the particular choice of the appeerence model and many feature engineering and feature learning techniques can be substituted here without the loss of generality. Our stochastic likelihood incorporates these histograms, by multiplying together the appearance probabilities for each image region $r \in R$. These probabilities, denoted $\vec{\theta_r}$, are smoothed by pseudo-counts $\epsilon$ drawn from a Gamma distribution. Let $Z_r$ be the per-region normalizing constant, and $I_{D_{(x,y)}}$ be the quantized pixel at coordinates $(x, y)$ in the input image. Then our likelihood model is:

$$P(I_D|I_R, \epsilon) = \prod_{r \in R} \prod_{x,y \text{ s.t. } I_R = r} \frac{\theta_r^{I_{D(x,y)}} + \epsilon}{Z_r}$$

Figure 5f shows appearance model histograms from one random training frame. Figure 5c shows the extremely noisy lane/non-lane classifications that result from the appearance model on its own, without our scene prior; accuracy is extremely low. Other, richer appearance models, such as Gaussian mixtures over RGB values (which could be either hand specified or learned), are compatible with our formulation; our simple, quantized model was chosen primarily for simplicity. We use the same generic Metropolis-Hastings strategy for inference in this problem as in our text application. Although deterministic search strategies for MAP inference could be developed for this particular program, it is less clear how to build a single deterministic search algorithm that could work on both of the generative probabilistic graphics programs we present.

In Table 1, we report the accuracy of our approach on one road dataset from the KITTI Vision Benchmark Suite [5]. To focus on accuracy in the face of visual variability, we do not exploit temporal correspondences. We test on every 5th frame for a total of 80. We report lane/non-lane accuracy results for maximum likelihood classification over 10 appearance models (from 10 randomly chosen training images), as well as for the single best appearance model from this set. We use 10 posterior samples per frame for both. For reference, we include the performance of a sophisticated bottom-up baseline system from [2]. This baseline system requires significant 3D a priori knowledge, including

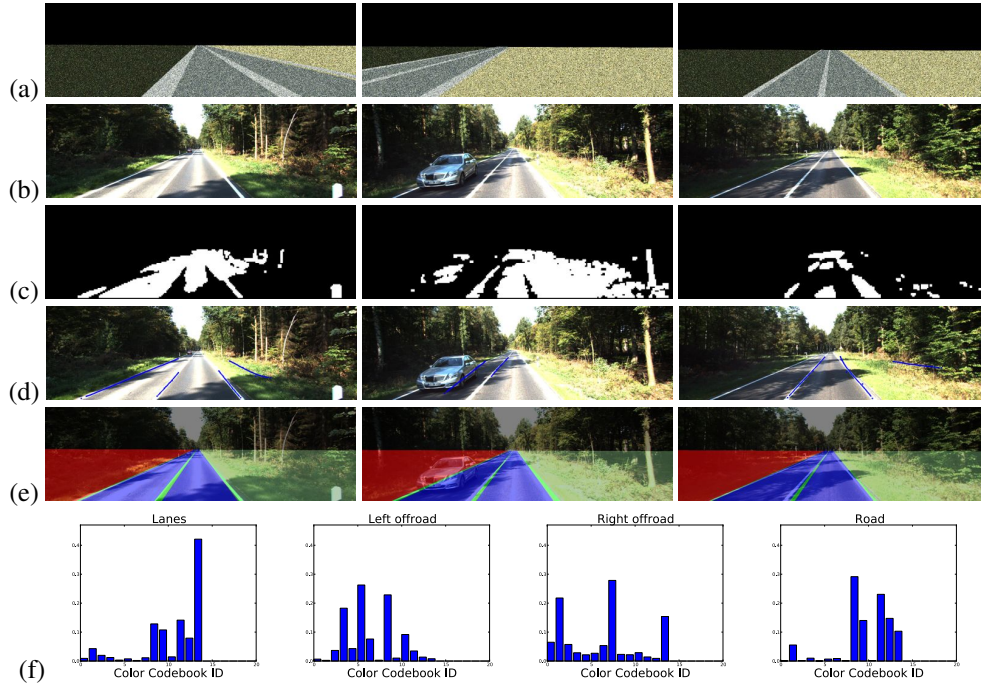

Figure 5: An illustration of generative probabilistic graphics for 3D road finding. **(a)** Renderings of random samples from our scene prior, showing the surface-based image segmentation induced by each sample. **(b)** Representative test frames from the KITTI dataset [5]. **(c)** Maximum likelihood lane/non-lane classification of the images from (b) based solely on the best-performing single-training-frame appearance model (ignoring latent geometry). Geometric constraints are clearly needed for reliable road finding. **(d)** Results from [2]. **(e)** Typical inference results from the proposed generative probabilistic graphics approach on the images from (b). **(f)** Appearance model histograms (over quantized RGB values) from the best-performing single-training-frame appearance model for all four region types: *lane*, *left offroad*, *right offroad* and *road*.

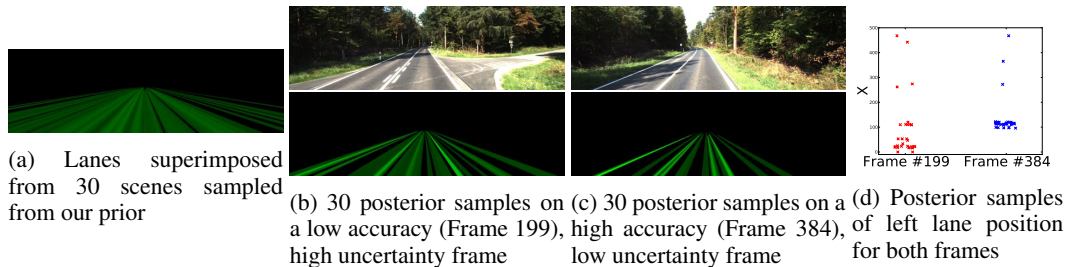

(a) Lanes superimposed from 30 scenes sampled from our prior

(b) 30 posterior samples on a low accuracy (Frame 199) high uncertainty frame

(c) 30 posterior samples on a high accuracy (Frame 384), low uncertainty frame

(d) Posterior samples of left lane position for both frames

Figure 6: Approximate Bayesian inference yields samples from a broad, multimodal scene posterior on a frame that violates our modeling assumptions (note the intersection), but reports less uncertainty on a frame more compatible with our model (with perceptually reasonable alternatives to the mode).

the intrinsic and extrinsic parameters of the camera, and a rough initial segmentation of each test image. In contrast, our approach has to infer these aspects of the scene from the image data. We also show some uncertainty estimates that result from approximate Bayesian inference in Figure 6. Our probabilistic graphics program for this problem requires under 20 lines of probabilistic code.

## 5 Discussion

We have shown that it is possible to write short probabilistic graphics programs that use simple 2D and 3D computer graphics techniques as the backbone for highly approximate generative models. Approximate Bayesian inference over the execution histories of these probabilistic graphics

```
ASSUME road_width (uniform_discrete 5 8) //arbitrary units
ASSUME road_height (uniform_discrete 70 150)
ASSUME lane_pos_x (uniform_continuous -1.0 1.0) //uncentered renderer
ASSUME lane_pos_y (uniform_continuous -5.0 0.0) //coordinate system
ASSUME lane_pos_z (uniform_continuous 1.0 3.5)
ASSUME lane_size (uniform_continuous 0.10 0.35)
ASSUME eps (gamma 1 1)
ASSUME theta_left (list 0.13 ... 0.03)
ASSUME theta_right (list 0.03 ... 0.02)
ASSUME theta_road (list 0.05 ... 0.07)
ASSUME theta_lane (list 0.01 ... 0.21)
ASSUME data (load_image "frame201.png")
ASSUME surfaces (render_surfaces lane_pos_x lane_pos_y lane_pos_z
  road_width road_height lane_size)
OBSERVE (incorporate_stochastic_likelihood theta_left theta_right
  theta_road theta_lane data surfaces eps) True
```

Figure 7: Source code for a generative probabilistic graphics program that infers 3D road models.

| Method | Accuracy |
|---|---|
| Aly et al [2] | 68.31% |
| GPGP (Best Single Appearance) | 64.56% |
| GPGP (Maximum Likelihood over Multiple Appearances) | 74.60% |

Table 1: Quantitative results for lane detection accuracy on one of the road datasets in the KITTI Vision Benchmark Suite [5]. See main text for details.

programs — automatically implemented via generic, single-variable Metropolis-Hastings transitions, using existing rendering libraries and simple likelihoods — then implements a new variation on analysis by synthesis [21]. We have also shown that this approach can yield accurate, globally consistent interpretations of real-world images, and can coherently report posterior uncertainty over latent scenes when appropriate. Our core contributions are the introduction of this conceptual framework and two initial demonstrations of its efficacy.

To scale our inference approach to handle more complex scenes, it will likely be important to consider more complex forms of automatic inference, beyond the single-variable Metropolis-Hastings proposals we currently use. For example, discriminatively trained proposals could help, and in fact could be trained based on forward executions of the probabilistic graphics program. Appearance models derived from modern image features and texture descriptors [14, 7, 11] — going beyond the simple quantizations we currently use — could also reduce the burden on inference and improve the generalizability of individual programs. It is important to note that the high dimensionality involved in probabilistic graphics programming does not necessarily mean inference (and even automatic inference) is impossible. For example, approximate inference in models with probabilities bounded away from 0 and 1 can sometimes be provably tractable via sampling techniques, with runtimes that depend on factors other than dimensionality [3]. Exploring the role of stochasticity in facilitating tractability is an important avenue for future work.

The most interesting potential of GPGP lies in bringing graphics representations and algorithms to bear on the hard modeling and inference problems in vision. For example, to avoid global re-rendering after each inference step, we need to represent and exploit the conditional independencies between latent scene elements and image regions. Inference in GPGP based on a z-buffer or a layered compositor could potentially do this. We hope the GPGP framework facilitates image analysis by Bayesian inversion of rich graphics algorithms for scene generation and image synthesis.

**Acknowledgments**
We are grateful to K. Bonawitz and E. Jonas for preliminary work on CAPTCHA breaking, and to S. Teller, B. Freeman, T. Adelson, M. James, M. Siegel and anonymous reviewers for helpful feedback and discussions. T. Kulkarni was graciously supported by the Henry E Singleton (1940) Fellowship. This research was supported by ONR award N000141310333, ARO MURI W911NF-13-1-2012, the DARPA UPSIDE program and a gift from Google.

## Footnotes

* The first two authors contributed equally to this work.

* (vkm, tejask, perov, jbt)@mit.edu — Project URL: http://probcomp.csail.mit.edu/gpgp/

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
