[Reviews · NeurIPS 2013]

Submitted by Assigned_Reviewer_3

This paper proposes a general method for solving image-based computer vision tasks using a generative probabilistic model that uses a graphics program to generate images. The method takes the standard Bayesian approach to frame the inference of the target variables, and uses Metropolis-Hastings to perform the inference. This framework is implemented for a CAPTCHA and a road-finding application, with favorable results reported for each one.

The primary contribution of this paper is a proposal for using graphics programs as a key element of a generative model for image-based tasks. While their claim that there are no previous real-world image interpretation frameworks that combine computer graphics among the other elements they list (last paragraph of Section 1) seems accurate, their proposed system does not seem to qualify as such a framework unless it's under a restricted interpretation. Specifically, the CAPTCHA problem is "real-world" in one sense, but CAPTCHAs are generated by a synthetic process and are therefore qualitatively a very different type of problem than such real-world problems as object recognition or detection in natural images. This distinction is important, because one of the main reasons that discriminative approaches have gained favor is due to their ability to deal with complex, noisy features found in natural images. The authors would presumably agree that their case is severely weakened if their framework is limited to synthetic images such as CAPTCHAs.

As for the road-finding problem, it certainly qualifies as a real-world problem and deals with natural images, but it is a strain to consider what they use to be a graphics program. The elements that the model generates are simply the properties (height, width, and lane size) of the lines that form the lanes of the road, and the "renderer" is simply an assignment of pixels to the regions delineated by these properties rather than an actual realistic rendering of the scene. This is perfectly reasonable, but it is no more of a computer graphics-based approach than the many previous approaches that align geometric elements to a scene (e.g., [3], [8], Hedau et al. ICCV '09, Lee et al. NIPS '10, Gupta et al. ECCV '10, and many similar projects by some of the authors of those papers). As such it should be seen as another work in this line of research, not a unique and novel direction that it would be if it were applied using a generative model that produced something closer to an actual image than geometric lines.

The authors point out that many other works require a good amount of application-specific engineering, which they do not require. While this is true, they also admit that such engineering would be a useful addition. In fact, in the Introduction section they point to [17, 20] as examples of methods that rely on custom inference strategies -- specifically, MCMC proposals that incorporate bottom-up cues -- in contrast to their more general approach, and then in the Discussion section they say that discriminatively trained proposals would likely be important to extend their approach to more complex scenes.

Novelty aside, the proposed framework does have attractive properties and the paper is well written, and the recent line of generative models for computer vision problems seems to be worth pursuing and has yielded promising results. While the probabilistic aspect is relatively straightforward, it is simple and easily adjusted according to the needs of the application (such as incorporating rotation), and the use of Metropolis-Hastings is elegant and general. The authors report favorable results on two interesting applications. It would have been better to compare the system's performance on the CAPTCHAs to further state-of-the-art approaches. Furthermore, while they compare directly to [1] on the road-finding application, it would be helpful to provide a means of comparison to the other results reported in the dataset's original paper ([4]).

As a technical note, it seems that the first equation should contain an integral over X, which is latent and does not appear on the left hand side.
Summary: While the proposed generative probabilistic approach is an attractive one, the novelty is far overstated. The authors report positive results in two applications which are enough to validate the approach in general, but not enough to warrant acceptance as an application paper.

Submitted by Assigned_Reviewer_4

The manuscript reports a system for image interpretation. In brief, an image is characterized by a latent representation which is a graphics program which, when executed, will generate the image (at least, approximately). To infer the parameters of the graphics program, the system performs Metropolis-Hastings steps.

Overall, I like this manuscript a lot. In some sense, the ideas are not new --- analysis by synthesis has been around for a long time. But the implementation of these ideas here is elegant. I like the notion of characterizing images as latent graphics programs. In some ways, I also like the idea of searching for good parameters of the graphics programs via Metropolis-Hastings steps (although, in many cases, this is likely to be very computationally expensive). In addition, I like the two applications in the manuscript (reading CAPTCHAs and finding roads in images). And the manuscript is generally well written (though I wish it contained more details).

My biggest concern is that the system might not scale well. It seems likely that the system will require an enormous number of Metropolis-Hastings steps when applied to large-scale real-world image understanding problems. I wish that the authors commented on this in the manuscript.
Summary: In summary, I really liked this manuscript a lot. Elegant and thoughtful implementation. Excellent applications of system to interesting data sets.

Submitted by Assigned_Reviewer_7

This paper builds a generative framework for analyzing images: The content of the a scene has some distribution. The content is the mapped to an image by an approximate renderer, and a probabilistic model compares this image to the actual data, under the control of a set of latent variables that effectively control the smoothing.

The paper is well written and easy to read. My expertise is more in Bayesian modeling, than computer vision, but the method presented in this paper appears to me to be a potentially significant step forward in computer vision. I believe the approach is novel, and the results presented here are promising. The authors also outline some very interesting future directions for this work, and I look forward to seeing more.

In particular, the two applications here seem like they can be summarized by a relatively simple feature space with this model. I wonder how much feature selection will affect the performance in more complex scenes.

One complaint is that the figure legends & axis labels were ridiculously small and hard to read.
Summary: This is a solid paper, and the approach to the problem appears to be novel and promising.
Author Feedback

Author rebuttal: We thank all the reviewers for their thoughtful comments, constructive criticism, and encouragement.

1. Reviewer 3 expresses concerns about novelty, especially considering geometric scene models from the recent vision literature, and the potential role of discriminative proposals.

We see our main contribution as an approach to generative modeling in vision, where solving a new task in a new domain requires specifying only domain-specific aspects of scene and image representation, but no domain-specific (or even domain-general) aspects of inference. That is, we do have to specify the hypothesis space of scenes (expressed as a probabilistic program) and an appropriate image representation (which can be viewed as a noisy observation model, or family of similarity metrics for approximate Bayesian computation). But we do not have to specify or implement an inference algorithm. Instead, generic, automatic inference for probabilistic programs is used to interpret images, and in fact can report posterior uncertainty about scene composition.

Regarding bottom-up proposals: our suggestion for scaling further was not to rely on custom engineering, but instead leverage probabilistic programming and ABC. For example, we could automatically build detectors from the generative model by training on sampled scene+image pairs, and integrate this into the underlying probabilistic programming system.

2. Reviewer 3 also expresses doubts about the presence of a meaningful renderer in our road scene application, and about the relevance of our CAPTCHA results for more typical vision problems, where the input image data is not generated by a computer program.

Our road scene model does include a full generative model for images, based on extremely rough, per-region color histograms. We chose to implement the likelihood calculation for this renderer using adaptive k-means quantization of colors for speed and simplicity (rather than e.g. an explicit Gaussian mixture model, per region, with a per-pixel mixture component, integrated out to yield a likelihood). If our paper is accepted we will include full road scene renderings (instead of just the surface geometry with arbitrary colors that was included in our submitted version) so readers can get a sense of how cartoonish an approximate, stochastic renderer can be and still support posterior inference.

We agree that naturalistic image settings are interesting, and have begun preliminary work on applying GPGP to reading degraded text from camera images, with distortions due to motion blur, out-of-plane rotation, warping of the underlying surface, etc. We think CAPTCHAs highlight interesting, hard aspects of scene parsing that are de-emphasized in typical object recognition tasks, such as the handling of extreme occlusion/object overlap, and complement our road scene application, which emphasizes variability in appearance against a comparatively simple hypothesis space of scenes.

3. Reviewer 4 points out that significantly more complex scenes could potentially result in very slow inference convergence. Understanding this aspect of scaling is a very important area for future work. For example, we have found that increasing the dimensionality in the CAPTCHA problem by increasing the number of letters by ~5x does not affect the number of MH steps needed for reliable convergence; instead, the difficulty seems to be determined by the degree of overlap of the letters.

4. Reviewer 7 observes that the image representations we used in our applications are simple --- raw pixels for CAPTCHAs, and highly quantized pixels for road scenes --- and that more complex scenes might benefit from (or perhaps even require) more sophisticated or carefully designed representations, analogously to the design of distance functions in more traditional ABC applications. This is an important issue to be explored in future work, e.g. by comparing typical generic image representations from vision (extracted from both a rendered image and the input data) to learned representations as well as simpler pixel-based ones.